# Estimation of Nitrogen Concentration in Winter Wheat Leaves Based on a Novel Spectral Index and Machine Learning Model

**DOI:** 10.3390/plants14172772

**Published:** 2025-09-04

**Authors:** Shihao Cui, Zhijun Li, Zijun Tang, Wei Zhang, Tao Sun, Yue Wu, Wanli Yang, Guofu Chen, Youzhen Xiang, Fucang Zhang

**Affiliations:** Key Laboratory of Agricultural Soil and Water Engineering in Arid and Semiarid Areas of Ministry of Education, Northwest A&F University, Yangling 712100, China; 2022012353@nwsuaf.edu.cn (S.C.); tangzijun@nwsuaf.edu.cn (Z.T.); 2021012221@nwsuaf.edu.cn (W.Z.); 2021050986@nwsuaf.edu.cn (T.S.); 2022012363@nwsuaf.edu.cn (Y.W.); 2022012387@nwsuaf.edu.cn (W.Y.); 2022012367@nwsuaf.edu.cn (G.C.); youzhenxiang@nwsuaf.edu.cn (Y.X.); zhangfc@nwsuaf.edu.cn (F.Z.)

**Keywords:** winter wheat, hyperspectral, spectral index, correlation matrix, leaf nitrogen concentration, machine learning

## Abstract

Assessing crop nitrogen status is crucial for optimizing fertilization strategies and promoting sustainable production. Although hyperspectral data offer significant advantages for monitoring subtle physiological changes in crops, accurately determining nitrogen status based on spectral information remains challenging. In this study, field experiments were conducted during the jointing stage of winter wheat on the Loess Plateau from 2018 to 2020. Concurrent measurements of leaf nitrogen concentration (LNC) and hyperspectral reflectance were collected to derive three types of spectral parameters: traditional vegetation indices, two-dimensional optimal spectral indices, and three-dimensional optimal spectral indices. Spectral parameters exhibiting a significant correlation with LNC (*p* < 0.05) were selected and combined as inputs for three machine learning models—extreme learning machine (ELM), back-propagation neural network (BPNN), and random forest (RF)—to develop LNC estimation models. The results demonstrated that, among the traditional indices, the Double Difference Index (DDn) showed the strongest correlation with LNC (r = 0.674). Within the multidimensional optimal indices, the differential three-dimensional scattering index (DTSI) exhibited the highest sensitivity to LNC (r = 0.721) at wavelength combinations of 833 nm, 755 nm, and 802 nm. Moreover, Model Input Combination 5 (comprising empirical indices plus three-dimensional optimal indices) further enhanced estimation accuracy. The RF model using Combination 5 achieved the best performance on the validation set (R^2^ = 0.827, RMSE = 2.803 mg g^−1^, MRE = 7.664%), significantly outperforming other model–input combinations. This study confirms the feasibility and high accuracy of winter wheat LNC inversion using novel multidimensional spectral indices and provides a new approach for real-time, non-destructive monitoring of nitrogen status in winter wheat.

## 1. Introduction

Winter wheat (*Triticum aestivum* L.) possesses unique physiological adaptations that confer irreplaceable value to global food security and agricultural systems [1]. As the predominant contributor to total wheat production worldwide (approximately 75%), winter wheat is a staple food for billions of people, directly processed into bread, noodles, and other food products, and also serves as an important feedstock, thereby underpinning food security on multiple fronts [2,3]. Its extensive root system stabilizes soil and reduces erosion during the wet winter–spring season, while promoting soil and water conservation and nutrient cycling in crop rotations [4,5]. In summary, winter wheat is an indispensable pillar for achieving sustainable, intensive cereal production globally.

Leaf nitrogen concentration (LNC) is a key physiological indicator reflecting the nitrogen nutrition status and photosynthetic capacity of winter wheat [6,7]. LNC directly influences leaf photosynthetic performance [8], carbon assimilation efficiency [9], and dry matter accumulation [10]. Its spatio-temporal dynamics provide precise insights into crop nitrogen demand, offering a scientific basis for optimizing nitrogen fertilizer management [11]. However, traditional field-based LNC assessment relies heavily on destructive sampling combined with laboratory chemical analyses (e.g., Kjeldahl or Dumas combustion methods) [12]. Although these methods are considered the gold standard, they suffer significant limitations in practice. First, destructive sampling damages plants and precludes continuous monitoring of the same individuals or specific sites, thereby losing the ability to track dynamic changes in nitrogen nutrition over time [13]. Second, the time lag inherent in laboratory analyses means that LNC results reflect past rather than real-time nitrogen status, risking missed critical windows for timely fertilization decisions [14].

In light of these limitations, hyperspectral remote sensing has emerged as a key solution for precise, non-destructive, and rapid diagnosis of nitrogen status in winter wheat [6,15]. By capturing continuous, narrow-band reflectance data from the crop canopy, this technique sensitively detects subtle spectral features induced by leaf biochemical constituents [16]. The high spectral resolution enables the development of robust inversion models for accurate LNC estimation [17]. Compared with conventional methods, hyperspectral remote sensing offers non-destructive sampling, near real-time monitoring, and large-scale coverage, making it possible to generate timely maps of crop nitrogen status [18].

Numerous studies have applied hyperspectral techniques to monitor leaf nitrogen concentration. For example, Wang et al. (2023) combined vegetation, color, and texture indices with hyperspectral parameters using machine learning to estimate nitrogen in rice stems and leaves, achieving R^2^ = 0.70 and RMSE = 3.19 mg g^−1^ [19]. Wan et al. (2022) employed transfer learning and hyperspectral analysis to evaluate LNC across different species datasets, reducing RMSE by 36.76% [20]. Duan et al. (2019) estimated total leaf nitrogen in winter wheat using canopy hyperspectral data and vertical nitrogen distribution, obtaining R^2^ > 0.60 [7]. Although these studies fully exploited spectral reflectance information, differences in growth environment and phenological stage introduce variability in physiological traits. To address this, many researchers have used correlation-matrix methods to select optimal spectral indices, thereby extracting more informative features and improving model accuracy. Sun et al. (2024) used a correlation matrix to construct optimal spectral indices for multi-scale soybean LNC estimation; coupling these indices with a random forest model yielded R^2^ = 0.856, RMSE = 0.551 mg g^−1^, and MRE = 10.405% on the validation set [21]. Li et al. (2018) constructed optimal spectral indices and applied a support vector machine- false discovery rate (SVM-FDR) regression model to estimate rapeseed LNC, achieving R^2^ = 0.828 and RPD (Ratio of Performance to Deviation) = 2.358 [22]. Although two-band indices exploit pairwise interactions and help curb redundancy, they are intrinsically constrained to dyadic combinations. In correlation-matrix screening over continuous hyperspectral reflectance, “optimal” indices are typically formed by retaining only the two wavelengths most correlated with the target trait, thereby discarding bands that carry weaker yet complementary information. This approach cannot capture higher-order (≥3-band) synergies or cross-region structures (e.g., pigment–structure–water couplings), which limits robustness across conditions [23]. In contrast, three-dimensional optimal spectral indices, which consider more complex interactions among spectral bands, hold promise for more reliable and accurate assessments. While three-dimensional indices have shown potential in related modeling domains [24], their application to crop nitrogen monitoring remains underexplored.

The jointing stage of winter wheat, marking the onset of stem elongation and floret primordium differentiation, represents a critical transition between vegetative and reproductive growth, with heightened sensitivity to water, nutrient, and light availability [23]. Accurate estimation of LNC during this phenological phase is therefore essential. To address these challenges, this study conducted controlled field experiments in typical winter wheat plots on the Loess Plateau from 2018 to 2020. We systematically acquired canopy hyperspectral reflectance and synchronous destructive-sampling measurements of leaf nitrogen concentration. Our objectives were (1) to construct three-dimensional optimal spectral indices (3D-OSIs)—adding dimensionality via a correlation-matrix approach—and quantify their sensitivity to winter wheat leaf nitrogen concentration (LNC), thereby overcoming the limitation of traditional two-band indices that capture only pairwise interactions; and (2) to develop an optimal, non-destructive LNC monitoring model by coupling 3D-OSIs with ensemble machine-learning algorithms. This study provides a new solution for real-time, high-precision monitoring of the nitrogen nutritional status of winter wheat.

## 2. Results

### 2.1. Correlation Analysis Between Spectral Parameters and Nitrogen Concentration in Winter Wheat Leaves

The correlation analysis between empirical spectral indices and winter wheat LNC is summarized in Table 1. All seven indices exhibited significant relationships with LNC (*p* < 0.05). Among them, the DDn showed the strongest correlation (r = 0.674). Accordingly, DDI, MNDSI, DDn, RDVI, CI, MTCI, and Gitelson2 were combined as Combine 1.

For the two-dimensional optimal spectral indices (2D-OSI), Table 2 and Figure 1 present their correlation with LNC. Each 2D-OSI correlated significantly with LNC (*p* < 0.05) and generally outperformed the empirical indices. The top performer was the TSI at bands 360 nm and 757 nm (r = 0.685). Consequently, SASI, NDSI, TSI, MSR, MNDI, RSI, and DSI were combined as Combine 2.

This study further developed four novel three-dimensional optimal spectral indices (3D-OSI). As shown in Table 3 and Figure 2, all 3D-OSI were significantly correlated with LNC (*p* < 0.05) and exceeded the performance of both 2D-OSI and empirical indices, with correlation coefficients surpassing 0.70. The differential three-dimensional scattering index (DTSI), based on wavelengths 833 nm, 755 nm, and 802 nm, achieved the highest correlation (r = 0.721). RTSI, DTSI, RDTSI, and RATSI were combined as Combine 3.

Figure 3 shows the correlation coefficients between the spectral indices. The results show that the correlation coefficients between most spectral indices are high.

### 2.2. Construction of a Model for Estimating Nitrogen Concentration in Winter Wheat Leaves

In this section, Combines 1–3 were recombined to form four new input sets: Combine 4 (Combine 1 + Combine 2), Combine 5 (Combine 1 + Combine 3), Combine 6 (Combine 2 + Combine 3), and Combine 7 (Combine 1 + Combine 2 + Combine 3). Each input set was used to train ELM, BPNN, and RF models for winter wheat LNC estimation, with results shown in Figure 4. When single-type indices were used (Combines 1–3), three-dimensional optimal indices (Combine 3) yielded the highest model accuracy, followed by two-dimensional indices (Combine 2), while empirical indices (Combine 1) produced the lowest accuracy. For multi-type inputs, Combine 5 (empirical + three-dimensional indices) delivered superior performance. Notably, the RF model with Combine 5 achieved the best validation-set results (R^2^ = 0.827, RMSE = 2.803 mg g^−1^, MRE = 7.664%), corresponding to a 17.2% increase in R^2^, a 22.3% reduction in RMSE, and a 32.6% reduction in MRE compared to the RF model using Combine 3.

## 3. Discussion

LNC is a key physiological indicator reflecting crop nitrogen nutrition status and photosynthetic capacity [25]. Hyperspectral remote sensing, with its continuous narrow bands and high spectral resolution, can sensitively capture subtle spectral signals induced by changes in leaf biochemical constituents, thereby meeting the requirements for spatio–temporal monitoring of nitrogen dynamics during the jointing stage of winter wheat [14].

Among the empirical spectral indices evaluated in this study, the DDn exhibited the highest correlation with LNC during the jointing stage. This finding suggests that the wavelength combination selected for DDn is highly sensitive to variations in leaf nitrogen content. Mechanistically, DDn enhances contrast between vegetation absorption and reflectance through normalization, effectively reducing interference from solar irradiance and background soil [26]. Moreover, red-edge absorption features are closely linked to chlorophyll content [27], and chlorophyll synthesis requires substantial nitrogen [28]. Thus, leaf nitrogen content is tightly coupled to chlorophyll concentration, and because nitrogen fertilization directly influences chlorophyll levels and photosynthetic efficiency, DDn—which captures chlorophyll-related spectral changes—most effectively differentiates nitrogen status and explains its superior performance in LNC estimation.

For two-dimensional optimal spectral indices, TSI at 360 nm and 757 nm achieved the highest correlation with LNC in Combine 2. These bands span the ultraviolet–visible and near-infrared regions: 360 nm lies in the UV region, often associated with epidermal UV-absorbing compounds [29], while 757 nm falls on the high-reflectance plateau of the NIR, which is highly sensitive to chlorophyll content and internal leaf structure [30,31]. TSI likely couples UV and NIR signals to reflect multiple physiological impacts of nitrogen: adequate nitrogen enhances chlorophyll content and boosts NIR reflectance, while nitrogen status also influences secondary metabolites and leaf structural characteristics, altering UV absorption features [32,33]. By combining 360 nm and 757 nm, TSI leverages both red-edge chlorophyll absorption and structural scattering responses [34], explaining its advantage over other two-band indices in sensitively monitoring chlorophyll and structural variations.

The novel three-dimensional optimal spectral index DTSI (833 nm, 755 nm, 802 nm) exhibited the highest correlation with LNC. Compared with two-band indices, tri-band indices capture more complex spectral interactions. The DTSI band combination spans the transition from the red-edge (755 nm) to the NIR plateau (802 nm, 833 nm). The 755 nm band approximates the classic red-edge position, while 802 nm and 833 nm reside in the high-reflectance NIR region [35]. Changes in nitrogen content manifest as characteristic reflectance shifts across these bands: higher leaf nitrogen typically accompanies elevated chlorophyll concentration, causing a steep rise in red-edge reflectance, and modifications to chlorophyll content and leaf structure also impact NIR reflectance [10]. By simultaneously considering interactions among three bands, DTSI integrates pigment- and structure-related spectral signals, fully exploiting intrinsic correlations in hyperspectral data, which explains its strongest association with LNC. The narrow 741–850 nm band is frequently used for nitrogen modeling because it is rich in key spectral features of leaf biochemical constituents. The 740–760 nm region corresponds to the third overtone of N–H and combination bands of C–H/C–N vibrations, indicating protein content, while the 810–850 nm region is closely associated with characteristic absorptions of water (O–H bonds) and cellulose. Since nitrogen levels influence leaf water status and structural compound synthesis, these indirect response features also effectively reflect nitrogen status. Moreover, nitrogen-sensitive absorption peaks are concentrated within this interval, so its selection precisely captures the core spectral responses indicative of leaf nitrogen nutrition.

In LNC estimation models, we compared ELM, BPNN, and RF approaches. When using Combine 5 (empirical + three-dimensional indices) as input, the RF model achieved the highest accuracy, outperforming the other models and demonstrating RF’s superiority in capturing complex nonlinear relationships between hyperspectral data and nitrogen [8]. Random forest’s ensemble of decision trees—each trained on random samples and random feature subsets—efficiently fits nonlinear patterns, and its out-of-bag (OOB) error estimation and voting mechanism inherently guard against overfitting [36]; similar advantages have been reported for RF in rice nitrogen estimation [37]. In contrast, BPNN and ELM, while capable of modeling nonlinearities, are more sensitive to hyperparameters and initial weights, prone to local minima, and lack built-in feature-importance analysis [16,38]. Thus, RF offers greater robustness and interpretability in this context.

Notably, the RF model with Combine 7 did not improve performance and showed a slight decline compared with the performance of the best combination (such as Combine 5) in the RF model. This likely reflects the high redundancy and multicollinearity present in hyperspectral data [39,40]. Combine 5 already encompasses rich spectral information from empirical and three-dimensional indices, and TSI overlaps in spectral features, adding limited novel information. Including highly correlated variables can introduce noise and increase model complexity without enhancing predictive power. As previous studies have shown, excessive redundant features in high-dimensional spectral data lead to overfitting and reduced generalization [41]; effective feature-selection methods must remove strongly collinear variables to improve model robustness [42]. Therefore, Combine 5—excluding TSI—achieved better predictions with a more parsimonious feature set, whereas Combine 7’s additional redundancy undermined accuracy.

In summary, this study has made preliminary progress in estimating leaf nitrogen concentration (LNC) using spectral indices, but several limitations remain and warrant further exploration. First, the interpretability of the models has not been fully exploited. Although correlation coefficients reveal the strength of linear relationships, they do not provide quantitative insights into the relative contributions of spectral features across different algorithms or their potential nonlinear interactions. Future work should incorporate interpretability methods such as SHAP [43,44] and permutation importance to visually demonstrate the specific roles of each index in model predictions and should leverage partial dependence plots to analyze their response patterns in depth. Second, the spatiotemporal diversity of the data must be enhanced. Our sample set was limited to a single season and location, which may not capture the effects of varying climate gradients, soil types, or growth stages on model performance. Future studies could collect data synchronously across multiple years and regions, and integrate multisource, multitemporal datasets by combining hyperspectral unmanned aerial vehicle (UAV) imagery with ground-based sensors to improve the generalizability and robustness of the models. Third, improving model generalization in small-sample settings remains a challenge. Beyond traditional cross-validation and regularization strategies, researchers could explore semi-supervised learning, transfer learning, or even meta-learning frameworks to transfer prior knowledge from other crops or similar regions to the target scenario. These approaches may reduce the reliance on large labeled datasets and enhance adaptability to new contexts. Additionally, implementing online learning and incremental update mechanisms could allow models to continuously refine themselves as new data become available during ongoing monitoring. Finally, the research should move closer to practical application. Building on the aforementioned improvements, a lightweight, real-time LNC diagnostic system could be developed and validated in field demonstrations to assess its effectiveness in guiding various management practices, such as nitrogen fertilization strategies and water management. By systematically integrating remote sensing technologies, machine learning interpretability techniques, and small-sample strategies, future work will provide more comprehensive and reliable technical support for crop nutrition monitoring and management in precision agriculture.

## 4. Materials and Methods

### 4.1. Research Area and Test Design

Field experiments in the 2018–2019 and 2019–2020 growing seasons were conducted at the Water-saving Station of the Key Laboratory of Agricultural Water and Soil Engineering in the Dryland Farming Region, Northwest A&F University, Yangling, China (34°18′ N, 108°24′ E; elevation 521 m). The site is characterized by a semi-humid, drought-prone climate, with an annual mean precipitation of 632 mm and a potential evapotranspiration of 1500 mm. A completely randomized block design with two replicates was employed, comprising 34 plots (7 m × 3 m each). The cultivar Xiaoyan 22 was sown at a rate of 180 kg ha^−1^ (sowing dates: 15 October 2018 and 15 October 2019; harvest dates: 25 June 2019 and 25 June 2020).

Five nitrogen (N) application rates were arranged: N0 (control, 0 kg N ha^−1^), N1 (100 kg N ha^−1^), N2 (160 kg N ha^−1^), N3 (220 kg N ha^−1^), and N4 (280 kg N ha^−1^), each combined with one of four fertilization regimes: conventional urea (URE), slow-release fertilizer (SRF), and two urea/SRF mixtures (UNS1: U/SRF = 3/7; UNS2: U/SRF = 2/8), yielding a total of 17 treatments. Basal phosphorus (120 kg P_2_O_5_ ha^−1^) and potassium (100 kg K_2_O ha^−1^) fertilizers were uniformly applied and incorporated into the 0–15 cm soil layer prior to sowing. Detailed field management and experimental procedures followed those described in [23].

### 4.2. Data Collection and Preprocessing

#### 4.2.1. Hyperspectral Data Acquisition

Canopy hyperspectral data of winter wheat were acquired during the jointing stage on 31 March 2019 and 3 April 2020 between 11:00 and 13:00 h, corresponding to a stable solar elevation angle under clear, windless conditions. A portable ASD FieldSpec 3 spectroradiometer (Analytical Spectral Devices, Boulder, CO, USA) was used to measure reflectance across the 350–1830 nm spectral range, with instrument settings and calibration procedures following [16]. In each plot, three representative 1 m × 1 m quadrats were selected, and nine consecutive spectral scans were collected per quadrat and corrected for the instrument’s field of view (take the winter wheat areas with the average growth of the whole plot, hyperspectral reflectance is collected directly above the plant to minimize soil interference). After removing outliers beyond ±3 standard deviations, the remaining spectra were averaged to yield the mean reflectance for each quadrat.

#### 4.2.2. Leaf Nitrogen Concentration Acquisition

Destructive sampling was conducted concurrently with hyperspectral measurements on the same days. Six uniformly vigorous winter wheat plants were selected per plot. At the jointing stage, before the flag leaf had fully emerged, the first fully expanded leaf on the main stem (typically the penultimate leaf) was harvested for chlorophyll and nitrogen analysis. Fresh leaf samples were immediately weighed to 0.001 g precision. Samples were then “killed” in a 105 °C forced-air oven for 30 min to halt enzyme activity, followed by drying at 75 °C to constant weight (approximately 48 h) to determine dry mass. Dried samples were ground, passed through a 40-mesh sieve, and 0.2000 ± 0.0005 g of powder was accurately weighed. Total nitrogen was determined by digesting the powder in concentrated sulfuric acid with catalyst at 420 °C until the solution was clear, followed by Kjeldahl distillation and titration [45].

#### 4.2.3. Spectral Data Preprocessing

To effectively eliminate background noise, baseline drift, and stray-light interference in the raw spectra, the Savitzky–Golay (SG) convolution smoothing algorithm was applied to the hyperspectral reflectance curves. A second-order polynomial was locally fitted over adjacent wavelength points within a nine-band window, preserving key spectral features while attenuating high-frequency noise. This procedure substantially enhanced the signal-to-noise ratio of the spectral data, thereby providing a robust foundation for subsequent spectral-parameter construction [21].

### 4.3. Selection and Construction of Spectral Parameters

Spectral indices serve as sensitive indicators of crop physiological status, and their construction strategies directly influence the accuracy of growth and nitrogen monitoring. In this study, we systematically integrated three classes of spectral-index frameworks:(1)Empirical Spectral Indices (SIs): Seven well-established indices (e.g., DDI, RDVI) with proven correlations to crop physiological and growth parameters were selected, calculated strictly according to references (Table 4).(2)Two-Dimensional Optimal Spectral Indices (2D-OSI): Optimal two-band combinations (Rᵢ, Rⱼ) were identified via a bivariate correlation-matrix approach, calculated strictly according to references (Table 5).(3)Three-Dimensional Optimal Spectral Indices (3D-OSI): The 3D-OSI is an extension of the 2D-OSI that introduces an additional dimension to better capture complex relationships among multiple spectral bands (Rᵢ, Rⱼ, R_k_). By explicitly modeling interactions among bands, 3D indices represent a broader range of spectral information and improve the sensitivity and accuracy of remote-sensing models, the formula used is “inspired” by common formulas such as NDSI (Table 6).

Within the usable spectral domain of 350–1830 nm, we employed a correlation-matrix procedure to screen optimal wavelength combinations. Using the modeling set as the basis, we exhaustively enumerated candidate indices for every wavelength pair/triad (Rᵢ, Rⱼ, R_k_) under generic operators (RSI/DSI/NDSI and RTSI/DTSI/RATSI/RDTSI), computed each index’s Pearson correlation coefficient (r) with leaf nitrogen concentration (LNC), and assembled an r–R matrix. All spectral index calculation results were obtained using MATLAB R2022a software (MathWorks, Inc. Natick, MA, USA).

### 4.4. Model Methods

Building on the previous section, we constructed several types of model input combinations. Specifically, Combine 1 includes all traditional empirical spectral indices from Table 1 that are significantly correlated with LNC (*p* < 0.05); Combine 2 includes all two-band optimal spectral indices (2D-OSIs) from Table 2 with significant correlations to LNC (*p* < 0.05); and Combine 3 includes all three-band optimal spectral indices (3D-OSIs) from Table 3 with significant correlations to LNC (*p* < 0.05). By extension, Combine 4 jointly inputs Combine 1 and Combine 2; Combine 5 jointly inputs Combine 1 and Combine 3; Combine 6 jointly inputs Combine 2 and Combine 3; and Combine 7 merges all indices from Table 1, Table 2 and Table 3—i.e., empirical + 2D-OSIs + 3D-OSIs—that are significantly correlated with LNC (*p* < 0.05) as the model’s input set.

A three-stage modeling framework was implemented to estimate winter wheat LNC. First, spectral indices significantly correlated with LNC (*p* < 0.05) were selected to form the input variable set. Second, three machine-learning models—extreme learning machine (ELM) [38], back-propagation neural network (BPNN) [52], and random forest (RF) [53]—were independently constructed. Finally, systematic parameter optimization was performed to identify the optimal architecture for each model. All running results of three machine-learning methods were obtained using MATLAB R2022a software (MathWorks, Inc. Natick, MA, USA).

(1)RF: Model convergence was assessed using out-of-bag (OOB) error. The number of trees was set to 100, and node splitting was based on the Gini impurity criterion [8].(2)ELM: A sigmoid activation function was used in the hidden layer. Input-to-hidden weights (aᵢ) and biases (bᵢ) were randomly initialized in the range [–1, 1]. Monte Carlo simulations (n = 50) determined that 1000 hidden-layer neurons balanced accuracy and computational efficiency [38].(3)BPNN: A single hidden layer with hyperbolic tangent (TANSIG) transfer functions was trained using the Levenberg–Marquardt algorithm (TrainLM). A grid search—varying hidden-layer neuron count from 15 to 120 in steps of 15—identified 15 neurons as optimal, yielding rapid convergence (R^2^ > 0.90) [23].

This fully parameterized modeling suite was then deployed to invert LNC and compare model performance.

### 4.5. Sample Set Division and Model Evaluation

A total of 68 valid samples were collected at the jointing stage. After outlier removal using Grubbs’ test (α = 0.05), 66 samples remained. Due to the moderate sample size (n = 66) after outlier removal, the dataset was partitioned into modeling and validation sets via stratified random sampling. This approach (modeling set for model building/tuning, validation set for final evaluation) is commonly adopted when sample size constraints preclude the use of a separate test set, ensuring robust performance assessment while maximizing data utility. The dataset was then partitioned by stratified random sampling into a modeling set (two-thirds of the data, n = 44) and a validation set (one-third, n = 22). The sample sizes and descriptive statistics of LNC for both subsets are shown in Figure 5. Model performance was evaluated using three metrics: coefficient of determination (R^2^), root mean square error (RMSE), and mean relative error (MRE). Origin Pro 2021 (OriginLab Corp., Northampton, MA, USA) was used to draw the figures.

## 5. Conclusions

This study developed an integrated framework that combines novel spectral indices with machine learning algorithms to estimate LNC in winter wheat during the jointing stage. By systematically comparing seven empirical indices, two-dimensional optimal spectral indices, and newly constructed three-dimensional indices, we identified DDn as the most sensitive empirical index (r = 0.674), TSI (360 nm/757 nm, r = 0.685) as the top two-dimensional index, and DTSI (833 nm, 755 nm, 802 nm, r = 0.721) as the best three-dimensional index. When these indices were combined and used as inputs for various machine learning models, the random forest (RF) model driven by the empirical + three-dimensional index combination (Combine 5) achieved the highest validation-set accuracy (R^2^ = 0.827, RMSE = 2.803 mg g^−1^, MRE = 7.66%), representing a 17.2% increase in R^2^, a 22.3% reduction in RMSE, and a 32.6% reduction in MRE compared to the RF model using only three-dimensional indices. This work demonstrates a high-precision, non-destructive approach for estimating winter wheat LNC and provides a practical pathway for intelligent monitoring of crop nitrogen status.

## Figures and Tables

**Figure 1 plants-14-02772-f001:**
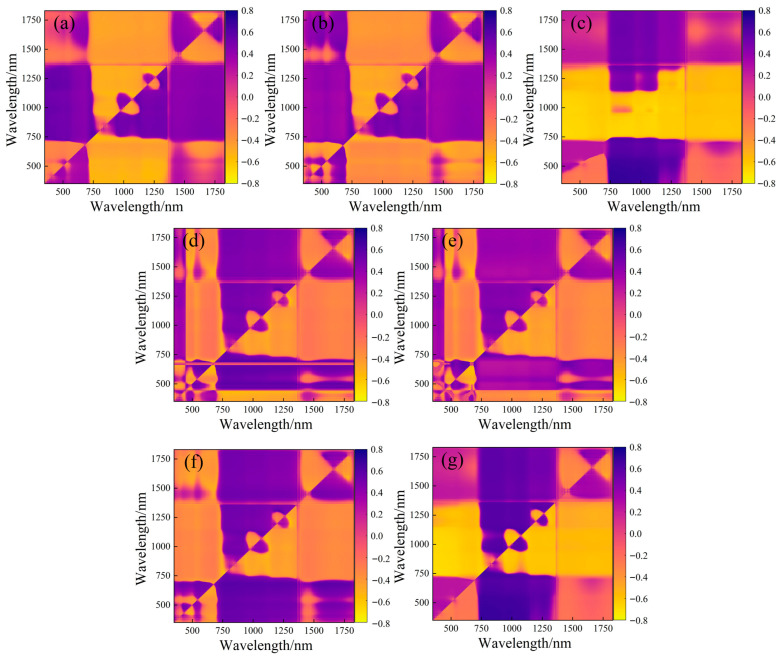
The correlation coefficient screening process of two-dimensional spectral index and winter wheat LNC. (**a**) SASI; (**b**) NDSI; (**c**) TSI; (**d**) mSR; (**e**) mNDI; (**f**) RSI; (**g**) DSI. Yellow to purple represents negative to positive correlation.

**Figure 2 plants-14-02772-f002:**
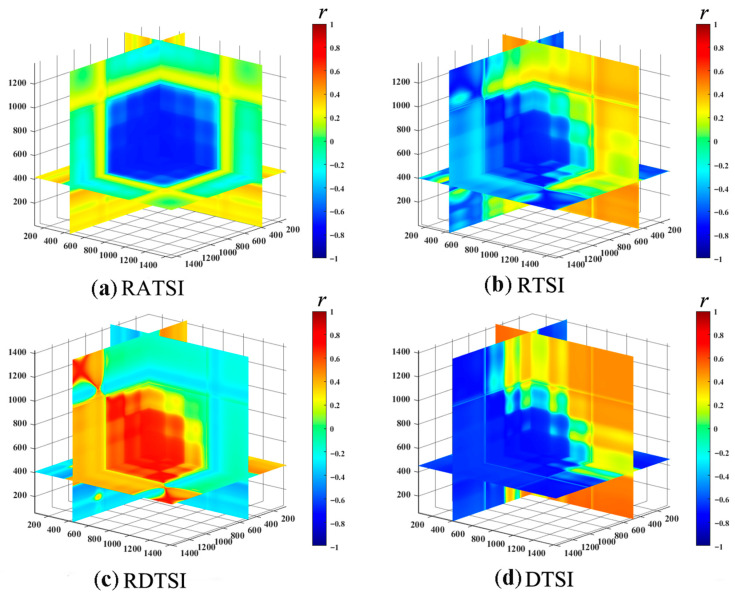
The correlation coefficient screening process of three-dimensional spectral index and winter wheat LNC. (**a**) RATSI; (**b**) RTSI; (**c**) RDTSI; (**d**) DTSI. Blue to red represents a negative to positive correlation.

**Figure 3 plants-14-02772-f003:**
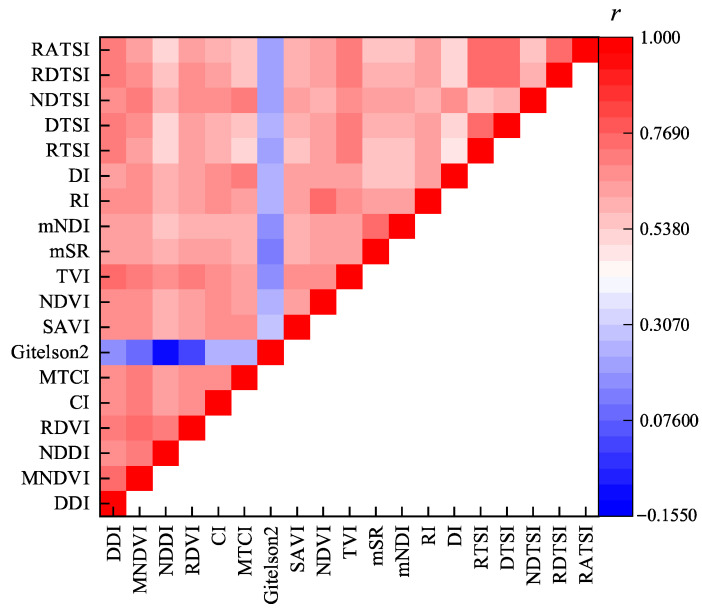
The correlation coefficient between each spectral index. Blue to red represents a negative to positive correlation.

**Figure 4 plants-14-02772-f004:**
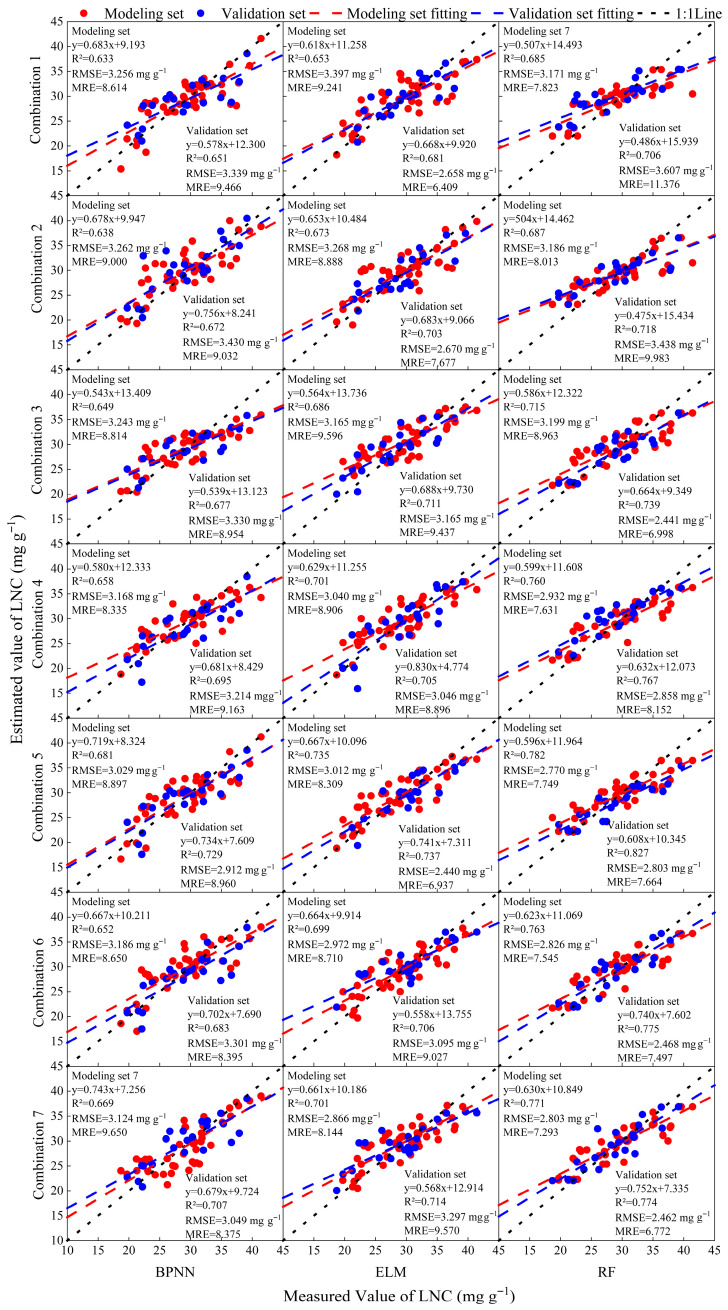
Construction of winter wheat LNC estimation model with different input combinations and machine learning models. The vertical and horizontal coordinates represent the combination of different model inputs and machine learning models.

**Figure 5 plants-14-02772-f005:**
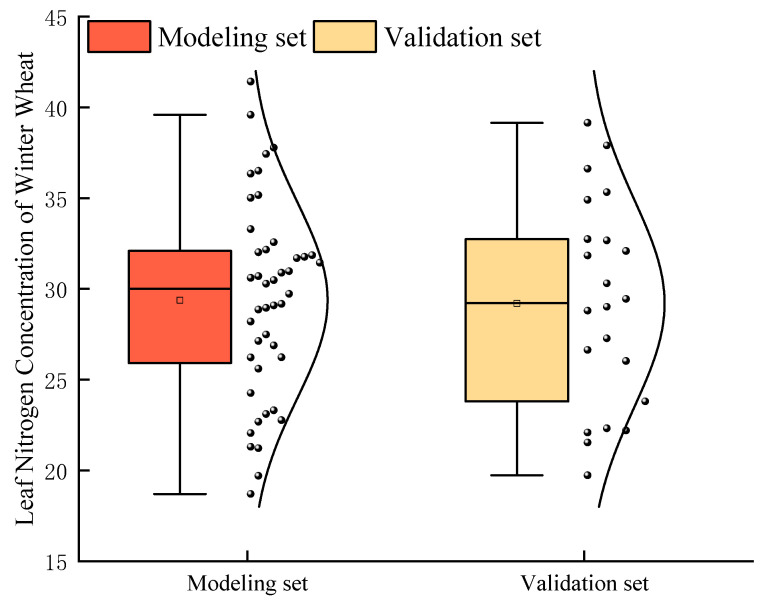
Statistics of winter wheat LNC (mg g^−1^) modeling set and validation set. The horizontal line in the box line diagram represents the median, and the white box represents the average value.

**Table 1 plants-14-02772-t001:** The correlation coefficient between empirical spectral index and LNC of winter wheat (* represents *p* < 0.05).

Spectral Indices	Correlation Coefficient
DDI	0.617 *
MNDSI	0.481 *
DDn	0.674 *
RDVI	0.592 *
CI	0.437 *
MTCI	0.538 *
Gitelson2	0.327 *

**Table 2 plants-14-02772-t002:** The correlation coefficient between the two-dimensional spectral index and the LNC of winter wheat (* indicates *p* < 0.05) and its band combination.

Spectral Indices	Correlation Coefficient	Position of Wavelength (i,j)/(nm)
SASI	0.628 *	(766, 761)
NDSI	0.629 *	(1206, 1307)
TSI	0.685 *	(360, 757)
mSR	0.647 *	(500, 505)
mNDI	0.658 *	(500, 505)
RSI	0.628 *	(1307, 1206)
DSI	0.545 *	(360, 757)

**Table 3 plants-14-02772-t003:** The correlation coefficient between the three-dimensional spectral index and the LNC of winter wheat (* indicates *p* < 0.05) and its band combination.

Spectral Indices	Correlation Coefficient	Position of Wavelength (i,j)/(nm)
RTSI	0.705 *	(783, 786, 750)
DTSI	0.721 *	(833, 755, 802)
RDTSI	0.569 *	(850, 792, 756)
RATSI	0.707 *	(741, 801, 765)

**Table 4 plants-14-02772-t004:** The used calculation formula of empirical spectral index and their references.

Spectral Indices	Formula	Reference
DDI (Dynamic dark index)	(749 nm−720 nm)−(701 nm−672 nm)	[46]
MNDSI (Modified normalized difference vegetation index)	$\frac{800\;nm-680\;nm}{800\;nm+680\;nm-2\times 445\;nm}$	[47]
DDn (Double difference index)	2×(710 nm−660 nm−760 nm)	[48]
RDVI (Renormalized difference vegetation index)	$\frac{800\;nm-670\;nm}{{\left(800\;nm+670\;nm\right)}^{0.5}}$	[49]
CI (Chlorophyll index)	675 nm $-\frac{690\;nm}{\;{683\;nm}^{2}}$	[46]
MTCI (MERIS terrestrial chlorophyll index)	$\frac{754\;nm-709\;nm}{709\;nm-681\;nm}$	[50]
Gitelson2 (Gitelson red-edge chlorophyll index 2)	$\frac{687\;nm-500\;nm}{750\;nm}$	[51]

Note: we used the original Le Maire et al. (2008) [48] definition of DDn: DDn(λ1,Δλ) = 2R(λ1) − R(λ1 − Δλ) − R(λ1 + Δλ), and we followed the canonical setting λ1 = 710 nm, Δλ = 50 nm, Δλ = 50 nm, i.e., 2R_710_ − R_660_ − R_760_.

**Table 5 plants-14-02772-t005:** The used calculation formula of two-dimensional optimal spectral index and their references. All indices were calculated from reflectance collected by spectroradiometer. The indices’ *i* and *j* represent arbitrary wavelength positions.

Spectral Indices	Formula	Reference
SASI (Soil-adjusted spectral index)	$\left(1+0.16\right)\frac{R_{i}-R_{j}}{R_{i}+R_{j}+0.16}$	[16]
NDSI (Normalized difference spectral index)	$R_{i}-R_{j}/R_{i}+R_{j}$	[16]
TSI (Triangular spectral index)	$0.5\times \left(120\times \left(R_{i}-R_{550}\right)-200\times \left(R_{j}-R_{550}\right)\right)$	[16]
mSR (Modified simple ratio)	$R_{i}-R_{455}/R_{j}-R_{455}$	[21]
mNDI (Modified normalized difference index)	$R_{i}-R_{j}/R_{i}+R_{j}-2R_{455}$	[21]
RSI (Ratio spectral index)	$R_{i}/R_{j}$	[21]
DSI (Difference spectral index)	$R_{i}$ − $R_{j}$	[21]

**Table 6 plants-14-02772-t006:** The used calculation formula of three-dimensional optimal spectral index. All indices were calculated from reflectance collected by spectroradiometer. The indices’ *i*, *j* and *k* represent arbitrary wavelength positions.

Spectral Indices	Formula
RTSI (Ratio triple spectral index)	$R_{i}/R_{j}/R_{k}$
DTSI (Difference triple spectral index)	$R_{i}-R_{j}-R_{k}$
RDTSI (Reciprocal difference triple spectral index)	${1/R}_{i}-1/R_{j}-1/R_{k}$
RATSI (Reciprocal additive triple spectral index)	${1/R}_{i}+1/R_{j}+1/R_{k}$

## Data Availability

The original contributions presented in this study are included in the article. Further inquiries can be directed to the corresponding authors.

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
