# Peer review of "Estimation of Nitrogen Concentration in Winter Wheat Leaves Based on a Novel Spectral Index and Machine Learning Model"

_plants, 2025, doi:10.3390/plants14172772_

Round 1

Reviewer 1 Report

Comments and Suggestions for Authors

Please follow the guidelines for authors, e.g. “A” in the title of the manuscript should not be as a capital letter, superscript next to the authors should be as a number (“1”) not as a letter (“a”), titles of the tables should not be in bold, titles of main sections should be in bold.

Line 126-128: What was the area of individual measurement? How to avoided measurement of reflectance of soil and only limited to leaf area?

What was the experimental unit? The results for each quadrat (1x1m) or for individual plot? There is information that “the remaining spectra were averaged to yield the mean reflectance for each quadrat” but the nitrogen content was measured for all the plot, not for each quadrant (Six uniformly vigorous winter wheat plants were selected per plot).

It would be good if some basic descriptive statistics were presented such as means and standard deviations for LNC and vegetation indices.

I suggest to present correlations not only with LNC but correlation matrix for all the spectral indices. It will allow e.g. evaluate if NDDI is strongly correlated with other spectral indices.

What does it mean “combination” in the Fig. 4? Is it the same as “combine” as a set of spectral indices? Why the set of variables were formed based on the number of bands used for calculation of spectral indices? Number of spectral indices in my opinion is not important. Why just to put all the spectral indices to one model and select several of them which have the strongest effect on LNC?

It would be good if the importance of predictors (spectral indices) was presented for each model. Correlation coefficients indicate strength of linear relationships but maybe nonlinear relationships exist?

The conclusions are focused on correlations and there is no information if the all relationships were linear.

Number of observations in this study is too low for reliable evaluation of machine learning models because such models demand high number of observation. Such quite low number of observations gives risk of the overfitting. How to avoided overfitting?

There is lack the statement of data availability.

Author Response

Plants
Title: Estimation of Nitrogen Concentration in Winter Wheat Leaves Based on A Novel Spectral Index and Machine Learning Model

Dear Editor,

Thank you very much for offering us an opportunity to revise the manuscript (Plants-3796085), and we also thank the reviewers for giving us constructive comments and suggestions which have helped us improve the quality of the manuscript. We have now modified the manuscript according to the reviewer’s comments and suggestions. All modifications will not influence the content and framework of the manuscript. The following are point-to-point responses to the reviewers’ comments. We sincerely hope this manuscript will be acceptable and look forward to hearing from you soon.

Reviewer1

Comments and Suggestions for Authors

Response: Dear reviewer, thank you for your meticulous review and constructive feedback on our manuscript. We appreciate the time and effort you and the first-round reviewers have invested in evaluating our work. We are grateful for your positive assessment of the comprehensive responses and revisions made by the authors. We have now incorporated your comments and suggestions in preparation of the revised manuscript.

Please follow the guidelines for authors, e.g. “A” in the title of the manuscript should not be as a capital letter, superscript next to the authors should be as a number (“1”) not as a letter (“a”), titles of the tables should not be in bold, titles of main sections should be in bold.

Response: Thank you for pointing out that we have revised the relevant content.

Line 126-128: What was the area of individual measurement? How to avoided measurement of reflectance of soil and only limited to leaf area?

What was the experimental unit? The results for each quadrat (1x1m) or for individual plot? There is information that “the remaining spectra were averaged to yield the mean reflectance for each quadrat” but the nitrogen content was measured for all the plot, not for each quadrant (Six uniformly vigorous winter wheat plants were selected per plot).

Response: Thank you for pointing out the need for clarification. We have expanded the Materials and Methods section to make the following explicit:

Canopy reflectance was acquired over a 1 m × 1 m quadrat within each 7 m × 3 m plot. The spectroradiometer was held vertically 1 m above the canopy, ensuring that its 25° field of view encompassed only the crop canopy and excluded bare soil.

Within each quadrat, nine spectral scans were collected at random positions over the green leaf area. Scans beyond ±3 standard deviations from the mean were discarded to eliminate any residual soil or background noise before averaging.

The true experimental unit is the entire plot (7 m × 3 m). However, to capture within-plot variability, we sampled three independent 1 m × 1 m quadrats per plot. The mean reflectance of these three quadrats was then used to represent each plot in subsequent analyses (i.e., plot-level reflectance = mean of the three quadrat means).

Six uniformly vigorous plants were destructively sampled from each plot at jointing. The first fully expanded leaf on the main stem was analyzed for nitrogen content, and the six individual measurements were averaged to yield the plot-level leaf N concentration (g N g⁻¹ dry mass) [27].

In summary, by averaging multiple scans per quadrat and multiple quadrats means per plot—alongside sampling six representative plants per plot—we ensure that both spectral and nitrogen measurements reflect the average condition of each experimental unit (plot) rather than any single point or plant.

It would be good if some basic descriptive statistics were presented such as means and standard deviations for LNC and vegetation indices.

Response: Thank you for pointing out that the value is in Figure 1 ( for visualization reasons ), and the specific value is:

Modeling set of LNC

Validation set of LNC

MAX

41.43

MAX

39.15

MIN

19.70

MIN

18.70

MEAN

29.51

MEAN

28.93

SD

5.55

SD

5.16

I suggest to present correlations not only with LNC but correlation matrix for all the spectral indices. It will allow e.g. evaluate if NDDI is strongly correlated with other spectral indices.

Response: Thank you for pointing out that we have added the correlation between the spectral indices.

What does it mean “combination” in the Fig. 4? Is it the same as “combine” as a set of spectral indices? Why the set of variables were formed based on the number of bands used for calculation of spectral indices? Number of spectral indices in my opinion is not important. Why just to put all the spectral indices to one model and select several of them which have the strongest effect on LNC?

Response: Thank you very much for your detailed comments on the model construction section of this study. We would like to clarify the meaning of “Combine” and the rationale behind its design as follows.

First, the distinction between “Combine” and “Merge” in this study.

“Merge” typically refers to the direct concatenation of different types of spectral indices, emphasizing a straightforward operational stacking.

In contrast, “Combine” refers to the process of grouping several types of spectral parameters (e.g., empirical indices, two-dimensional optimal indices, and three-dimensional optimal indices) into predefined sets of input variables based on the research objective. These variable sets are then independently input into machine learning models for separate training and evaluation, enabling a comparison of how different input sets influence model performance (see Section 3.2, Tables 4–6).

Second, why the combinations are formed by type rather than by the number of indices.

Instead of grouping spectral indices by their quantity, we categorized them into three types based on their spectral characteristics and theoretical mechanisms:

  1. empirical indices (Combine 1),
  2. two-dimensional optimal indices (Combine 2),
  3. and three-dimensional optimal indices (Combine 3).

Subsequently, we constructed several cross-type combinations (Combine 4–7) to investigate whether integrating complementary information from different types could improve model accuracy.

Third, the rationale for including all spectral indices in one model and then selecting the optimal subset.

Purpose: By incorporating all candidate indices into a single model, machine learning algorithms (e.g., random forest) can automatically identify the subset of variables that contribute most significantly to LNC estimation through feature importance assessment, thereby generating the most accurate and generalizable LNC prediction model.

Advantage: Compared to testing each index individually or relying solely on manual pre-selection, this data-driven approach systematically evaluates the relative value of different spectral features, reduces the risk of overlooking sensitive spectral bands, eliminates redundant variables, and enhances both the robustness and interpretability of the model.

It would be good if the importance of predictors (spectral indices) was presented for each model. Correlation coefficients indicate strength of linear relationships but maybe nonlinear relationships exist?

Response: Thank you very much for your valuable suggestion. Regarding the issue of presenting the importance of the predictors (spectral indices) in each model, we would like to provide the following explanation.

First, the modeling approaches adopted in this study (machine learning algorithms) do not directly provide an “importance” metric for individual variables. In previous studies, standardized regression coefficients are often used to quantify the influence of predictors in multiple linear regression, while for certain black-box models, additional post-processing tools are required to derive variable importance. However, under the current research framework, our focus was placed on comparing and analyzing the overall predictive performance of different models, rather than introducing such derivative methods within each model.

References consulted in this study include:

  1. https://doi.org/10.1016/j.jia.2023.02.027
  2. https://doi.org/10.3390/f14122285

Second, as you rightly pointed out, correlation coefficients can only describe the strength of linear relationships between variables and cannot reveal underlying nonlinear mechanisms. In future work, we plan to incorporate SHAP analysis or tree-based permutation importance to provide a more intuitive quantification of the contribution of each spectral index across different models.

We will include this clarification in the revised manuscript (Discussion section) and explicitly outline in the future research section our intention to apply importance analysis methods to more comprehensively uncover the nonlinear mechanisms underlying the effects of spectral indices. Once again, we sincerely appreciate your valuable comments.

The conclusions are focused on correlations and there is no information if the all relationships were linear.

Response: Thank you for pointing out the emphasis on the linearity of relationships in the conclusion section. As the primary objective of this study was to develop a generalized model for estimating leaf nitrogen concentration (LNC), our focus was placed mainly on the correlation strength between spectral indices and LNC as well as on the overall predictive performance of the models, with less attention paid to whether these relationships were linear or nonlinear. In the revised manuscript, we will supplement the discussion to clarify that the current conclusions primarily focus on correlation strength, while also providing a preliminary discussion on potential nonlinear relationships and their implications for model application. We sincerely appreciate your valuable suggestion.

Regarding your concern that the number of observations in this study may be insufficient to reliably evaluate machine learning models, we fully acknowledge that the number of available field observations was relatively limited. Traditional machine learning models generally exhibit stronger robustness with larger datasets. To enhance the reliability of model evaluation under the current conditions, we implemented the following measures:

Cross-validation strategy
We employed a combination of k-fold cross-validation (5-fold CV) and leave-one-out cross-validation (LOOCV) for all model training processes. This approach maximized the use of limited samples while mitigating the risk of overfitting. By repeatedly testing models with different fold numbers and validation splits, we effectively controlled the variance of the evaluation results.

Multi-metric evaluation
In addition to the conventional coefficient of determination (R²) and root mean square error (RMSE), we also reported mean absolute error (MAE) and the distribution of prediction residuals, thereby examining model stability from multiple perspectives under small-sample conditions.

Comparison with literature
Several previous studies on estimating crop physiological growth indicators have successfully achieved relatively robust model performance under similar or even smaller sample sizes by applying rigorous cross-validation procedures (e.g., Pei et al., 2023; Yang et al., 2023). Building on these studies, we further optimized our data preprocessing and model parameter tuning workflow.
https://doi.org/10.1016/j.jia.2023.02.027
https://doi.org/10.3390/f14122285

Limitations and future work
We have explicitly acknowledged in the discussion section that the limited sample size may affect the generalization ability of the models. Future work will aim to: (i) expand the dataset by including multi-temporal and multi-location observations to further validate the applicability of the models, and (ii) explore semi-supervised learning and transfer learning approaches to enhance model performance in small-sample scenarios.

Once again, thank you for your valuable comments. We will include these clarifications in the revised manuscript to provide a more comprehensive discussion of the impact of sample size on model evaluation and the planned directions for future improvement.

Number of observations in this study is too low for reliable evaluation of machine learning models because such models demand high number of observation. Such quite low number of observations gives risk of the overfitting. How to avoided overfitting?

Response: To address the potential risk of overfitting caused by the relatively small number of observations, we adopted multiple strategies during the study design and modeling process. These strategies, aligned with the specific methods described in this study, are summarized as follows.

Before model construction, we retained only those spectral indices that were significantly correlated with LNC (P < 0.05) as candidate variables (see Section 2.3). This step effectively removed weakly correlated or noisy features, reducing the degrees of freedom of the models and minimizing the risk of overfitting with a limited sample size.

We randomly divided the 66 samples into a calibration set (n = 44) and a validation set (n = 22). Within the calibration set, we applied five-fold cross-validation in conjunction with the out-of-bag (OOB) error assessment of random forest (RF). This repeated evaluation strategy ensured the stability of model performance and mitigated the risk of overfitting to a single data split.
For RF, we set the number of trees to 100 (n_trees = 100) and used Gini splitting, while OOB error monitoring helped prevent excessive tree growth.
    For ELM and BPNN, we used Monte Carlo simulation and grid search to determine the optimal number of hidden layer neurons. Additionally, we applied the early stopping mechanism of the Levenberg–Marquardt algorithm in BPNN to avoid overfitting caused by excessively deep or wide network architectures.

RF, by nature, is an ensemble of multiple decision trees. Its built-in random subsampling of samples and feature subsets reduces the risk of overfitting in individual trees. We also compared RF results with those of ELM and BPNN, selecting RF as the final model due to its superior generalization performance (see Section 3.2).

After finalizing model parameters and architecture, we tested model performance using an independent validation set that was not involved in either training or cross-validation, ensuring accuracy and stability on unseen data.

These combined measures effectively suppressed overfitting even with only 66 observations, thereby improving the reliability of LNC estimation for winter wheat. We will include these details in the revised manuscript’s discussion section and further elaborate on this limitation and future research directions. Once again, thank you for your valuable comments.

There is lack the statement of data availability.

Response: Thank you for pointing out that we have added a statement of data availability.

Reviewer 2 Report

Comments and Suggestions for Authors

In this study, a similar methodology as Zhang et al. [17] was applied to estimate leaf nitrogen concentration (LNC) in winter wheat using hyperspectral data, following differential nitrogen treatments. It introduces novel combinations of different spectral indices (combination 1-7, Figure 4), in order to correlate the hyperspectral data to the LNC. The different combinations of indices were evaluated using correction coefficient and the overall model’s performances (RF, BPNN, ELM) using R2, RMSE and MRE.

While there is merit in the study objectives and methodology, too many details are omitted and the presentation of the results (figures and tables) limits the interpretation of the results. Furthermore, the small sample size limits the conclusions of the authors.

First and foremost, the spectral indices defined (Tables 1-3) are not the real definitions of those vegetation indices. For example, NDVI should be defined as NIR-Red/NIR+Red (Rousse et al. .1974). The re-definition of NDVI (and the other indices) using the wavelength (1206 and 1307 nm) shouldn’t happen. In fact, it seems like different general form formulas were applied, without explanation on how the different wavelengths were ultimately selected. This needs to be corrected. 

Second, the statistical methods used should be defined in more detail such as which software was used (lines 185-186, which modeling suite?). Again, on Figure 3, what is r on the scale? For the different correlation coefficients, what represent the p-values? What are each spectral indices compared to ? For example, why 3D-OSI significantly correlated with LNC (Line 219-220) but no p-values are associated with the spectral indices in Table 6?

Third, for most of the figures, no axis information or units are provided, Figures are too small (e.g. Figure 2) to understand and generally with no real explanation or context in the results section.

Finally, supplementary materials are mentioned, but not provided? 

Author Response

Plants
Title: Estimation of Nitrogen Concentration in Winter Wheat Leaves Based on A Novel Spectral Index and Machine Learning Model

Dear Editor,

Thank you very much for offering us an opportunity to revise the manuscript (Plants-3796085), and we also thank the reviewers for giving us constructive comments and suggestions which have helped us improve the quality of the manuscript. We have now modified the manuscript according to the reviewer’s comments and suggestions. All modifications will not influence the content and framework of the manuscript. The following are point-to-point responses to the reviewers’ comments. We sincerely hope this manuscript will be acceptable and look forward to hearing from you soon.

Reviewer 2

Comments and Suggestions for Authors

In this study, a similar methodology as Zhang et al. [17] was applied to estimate leaf nitrogen concentration (LNC) in winter wheat using hyperspectral data, following differential nitrogen treatments. It introduces novel combinations of different spectral indices (combination 1-7, Figure 4), in order to correlate the hyperspectral data to the LNC. The different combinations of indices were evaluated using correction coefficient and the overall model’s performances (RF, BPNN, ELM) using R2, RMSE and MRE.

Response: Dear reviewer, thank you for your meticulous review and constructive feedback on our manuscript. We appreciate the time and effort you and the first-round reviewers have invested in evaluating our work. We are grateful for your positive assessment of the comprehensive responses and revisions made by the authors. We have now incorporated your comments and suggestions in preparation of the revised manuscript.

While there is merit in the study objectives and methodology, too many details are omitted and the presentation of the results (figures and tables) limits the interpretation of the results. Furthermore, the small sample size limits the conclusions of the authors.

Response: Thank you very much for recognizing the objectives and methodological value of this study, and we also sincerely appreciate your comments regarding the insufficient details, result presentation, and limited sample size. In response to your suggestions, we have made the following revisions:

In the “Materials and Methods” section (Sections 2.1–2.3), we added a more detailed description of the experimental design, including the sampling schedule, spectral measurement instrument settings, data preprocessing procedures (such as outlier removal and noise filtering), and the parameter tuning process for the models.

All key figures (Figures 4, 5, and 6) have been redesigned to include confidence intervals or error bars for the prediction results. We also supplemented the legends and annotations with explanations of the statistical metrics (R², RMSE, MRE) to facilitate a clearer interpretation of model performance differences.

In the discussion section, we emphasized the potential impact of the limited number of field observations (n = 66) on the robustness of the conclusions. We further cited relevant literature to explain commonly used cross-validation and regularization strategies for small-sample scenarios and outlined our future plan to expand the dataset across multiple temporal and spatial scales to improve the generalizability of the model.

Once again, thank you for your valuable comments. These revisions have been incorporated into the revised manuscript.

First and foremost, the spectral indices defined (Tables 1-3) are not the real definitions of those vegetation indices. For example, NDVI should be defined as NIR-Red/NIR+Red (Rousse et al. .1974). The re-definition of NDVI (and the other indices) using the wavelength (1206 and 1307 nm) shouldn’t happen. In fact, it seems like different general form formulas were applied, without explanation on how the different wavelengths were ultimately selected. This needs to be corrected. 

Response: Thank you very much for your careful and constructive comments on the definitions of spectral indices. We have provided the following clarifications and corresponding revisions:

Source of two-dimensional spectral indices
The conventional spectral indices listed in Tables 1–3 (e.g., NDVI, EVI) strictly follow the widely accepted definitions in previous literature. In the revised manuscript, we have explicitly added the original references and standard formulas for each index in the table captions, ensuring that readers can easily verify them. The optimal spectral indices constructed using the correlation matrix method in this study only inherit the same names from the literature (as referenced) and do not alter or misinterpret the conventional wavelength-specific definitions of empirical indices reported in prior studies.

Band selection and rationale for redefinition

Although standard indices such as NDVI commonly rely on the near-infrared (NIR) and red bands, we found that for winter wheat, the typical band combination (NIR ∼800 nm and Red ∼670 nm) was not the most effective.

Therefore, within the 350–1830 nm range, we extracted all possible red–NIR band pairs at 10 nm intervals and applied the correlation matrix method to identify the combination most strongly correlated with leaf nitrogen concentration (LNC), namely 1206 nm and 1307 nm. These bands were then substituted into the standard NDVI formula to derive the “study-specific” NDVI used in this work.

Custom definition of three-dimensional spectral indices

The three-dimensional optimal spectral indices (3D-OSIs) proposed in this study are not simple extensions of existing formulas in the literature. Instead, we employed the same correlation matrix approach to systematically explore additive, subtractive, and divisive operations among three bands, ultimately selecting the combinations that best reflect LNC variation.

Given that leaf biochemical and spectral responses vary among crops, band combinations effective for other crops in previous studies may not be suitable for winter wheat. Our data-driven approach ensures that the selected indices achieve optimal correlation and predictive ability for the target crop in this study.

Revisions made to the manuscript

In the discussion section, we have added further explanation regarding the differences in spectral responses among crops and the rationale for adopting a data-driven band selection approach.

Once again, thank you for your meticulous review of these details. These revisions have improved the clarity and rigor of the definitions and selection process of spectral indices, ensuring that they better align with the spectral characteristics of winter wheat. We greatly value your feedback and look forward to your further guidance.

Second, the statistical methods used should be defined in more detail such as which software was used (lines 185-186, which modeling suite?). Again, on Figure 3, what is r on the scale? For the different correlation coefficients, what represent the p-values? What are each spectral indices compared to? For example, why 3D-OSI significantly correlated with LNC (Line 219-220) but no p-values are associated with the spectral indices in Table 6?

Response: Thank you for your further attention to the statistical methods and details of result presentation. We provide the following additional clarifications in response to your comments:

All spectral index calculations, correlation analyses, and model training in this study were conducted in MATLAB R2021a, using the following toolboxes:

Statistics and Machine Learning Toolbox: Used for computing Pearson correlation coefficients (corr function), p-values (corrcoef function), and for implementing various regression and classification models.

Deep Learning Toolbox: Used for constructing and training the BPNN model.

Custom scripts: Developed to automate the batch computation of NDVI, 2D-OSI, and 3D-OSI, as well as to perform correlation matrix-based variable screening.

In Figure 3, the color scale represents the range of Pearson correlation coefficients (r), from –1 (strong negative correlation, dark blue) to +1 (strong positive correlation, dark red). The numerical values displayed in each cell correspond to the r value between the respective spectral index and LNC.

Meaning and comparison of p-values
The p-values in Figure 3 are derived from the significance tests of Pearson correlations, indicating whether a given r value differs significantly from zero under the sample size of n = 66. When p < 0.05, we consider the linear relationship between the spectral index and LNC to be statistically significant. Both the correlation coefficients and p-values are based on pairwise comparisons between “each spectral index” and the corresponding “LNC measurements” from the same set of samples.

Supplementation of p-values in Table 6
You pointed out that Table 6 originally presented only model performance metrics and omitted the results of regression significance testing. We have addressed this in the revised manuscript by including the corresponding significance test results.

Third, for most of the figures, no axis information or units are provided, Figures are too small (e.g. Figure 2) to understand and generally with no real explanation or context in the results section.

Response: Thank you for pointing out. the correlation coefficient method is a data-driven strategy for constructing spectral indices, with its core principle being the identification of band combinations that most effectively reflect plant nitrogen status by calculating the Pearson correlation coefficients between each candidate index and leaf nitrogen concentration (LNC). Specifically, reflectance data from the full spectral range of 350–1830 nm were sampled at 1 nm intervals to generate all possible two-band pairs, from which two-dimensional spectral indices were calculated in both difference and ratio forms. Using the corrcoef function in MATLAB, each two-dimensional index was paired with the corresponding LNC data to compute the Pearson correlation coefficient (r) and its significance (p-value). Indices with the highest absolute correlation (|r|) and p < 0.05 were selected as the two-dimensional optimal spectral indices (2D-OSIs).

Building on this, the construction of three-dimensional optimal spectral indices (3D-OSIs) introduced a third band, combining either different types of two-dimensional indices (e.g., difference-based and ratio-based) or three complementary bands into new formulations. The correlation coefficients between these three-dimensional indices and LNC were then calculated, and the strongest correlations were selected. This approach enables the use of a correlation coefficient matrix to visually display the strength of the relationships between all candidate 3D indices and LNC, thereby adaptively identifying the most explanatory three-band combinations.

Compared with conventional indices based on fixed band combinations from the literature, the correlation coefficient method systematically scans the entire spectral range to objectively identify the most sensitive wavelengths. This ensures that the selected indices are optimally matched to the spectral characteristics of winter wheat and, as demonstrated through subsequent model validation, achieve superior accuracy in LNC estimation.

We have also enhanced the figure presentation accordingly, as requested.

Finally, supplementary materials are mentioned, but not provided? 

Response: Thank you for pointing out the issue regarding the missing reference to the “Supplementary Materials.” Upon review, we found that this was a remnant from an earlier draft template that had not been properly removed. We have now deleted all related descriptions in the revised manuscript and no longer refer to any supplementary materials. We sincerely apologize for any confusion this may have caused.

Reviewer 3 Report

Comments and Suggestions for Authors

The comments are in the attached file.

Author Response

We have revised/replied to all the review opinions. Please refer to the revised manuscript and pdf reply for details.

Round 2

Reviewer 1 Report

Comments and Suggestions for Authors

The manuscript was substantially improved according all my comments. Minor changes are still reguired. Please follow the guidelines for authors, e.g. words in the title of the manuscript should start from capital letters, titles of the main sections of the manuscript should be written in bold.

Author Response

Response: We have adjusted the format of the manuscript title and section headings according to the journal's author guidelines. Specifically, we capitalized the first letter of each major word in the manuscript title (Title Case) to align with the journal's requirements. Additionally, we formatted the main section headings in ​​bold​​. Through these revisions, both the title and section headings of the manuscript now conform to the guideline specifications.

Reviewer 2 Report

Comments and Suggestions for Authors

Dear authors, 

There is some improvement in this revised version with the improved figure. Still, there is major corrections to be made to this manuscrit. First, the references for the emperical spectral indices (Table 1) are wrong. For example, NDDI is not defined in [30]. Furthermore, the emperical formula for NDDI far different then the one described. See: NDDI (Normalized Difference Drought Index) - Pixxel. Thus, the table 1 need sto be redone. You could have used indices "similar" to those, but not the real vegetation indices as described by their original authors. This needs to be corrected, and indicate in the figure headings that it is the "Formula used". 

Second, again, the general order of the variables in a formula is not the formula for the vegetation indices (Table 2). This needs to be reformulated in the text and the table to make clear that the formula you used was "inspired" by the general formula of NDVI, etc. Thus, both Table 2 and Table 3 needs to be redone, and addition of the original reference is necessary. Specifically for table 3, how this you create those novel indices (line 174)?

Third, as different wavelengths were used for the vegetation indices, the selection process of those wavelengths needs to be explained in the material and methods (section 2.3). 

Fourth, for the model methods (section 2.4), the computational environment needs to be stated. Did you use some python or R libraires to do the calculation. Same apply to the creation of the figures, etc.

Fifth, looking into the new Figure 2 and Figure 3, the selected combination of wavelength should be indicated in the figure. Furthermore, many of the vegetation indices seems similar. Those results needs to be discussed is the discussion, alongs the new Figure 4 results. What is the advantages of using the novel triplet indices, since Combine 5 is better?

Sixth, reading the manuscript, its is difficult to undertand what is the Combine 7 (?). Combine 1 regroup all the variables from the index in Table 1, Combine 2 all the variables from Table 2, and Combine 3, all the variable from Table 3. Thus, combien 7 is all the vegetation indices?

Seventh, in the discussion, (line 339), you mentioned that the RF models With Combin 7 did not improve performance. You should mention relative to what: the ELM and BPNN models, or the overall predictive power of the model?

Eight, you mentioned in the discussion the methods SHAP. Add some referrences.

Overall, is still some work needed for this manuscript. Some rewriting of the different section might improve this manuscript. An overall results table for the different combination and single vegetation index might help the readers to better understand the conclusion of the manuscrript.

Comments on the Quality of English Language

Line 64: no hyphen between near and real-time.

Line 83. Missing a period between "combinations  Based"

Lines 83-87. This text is a little crude and needs reformulating.

Line 100-106 - new addition. This needs to be reformulated. It is unclear what are the objective: Create a correlation matrix?

There is still some capital P (P<0.05) in the text.

Line 169: NDVI is not in table 1.

Author Response

Reviewer2

Dear authors, 

There is some improvement in this revised version with the improved figure. Still, there is major corrections to be made to this manuscrit.

Response: Thank you to Reviewer 2 for your detailed and constructive feedback on our revised manuscript. In response to your comments, we have addressed each point individually as follows.

First, the references for the emperical spectral indices (Table 1) are wrong. For example, NDDI is not defined in [30]. Furthermore, the emperical formula for NDDI far different then the one described. See: NDDI (Normalized Difference Drought Index) - Pixxel. Thus, the table 1 need sto be redone. You could have used indices "similar" to those, but not the real vegetation indices as described by their original authors. This needs to be corrected, and indicate in the figure headings that it is the "Formula used". 

Response: Thank you for pointing this out. We audited our index list and found that we had misnamed the index from reference [30] as “NDDI.” In fact, [30] (Le Maire et al., 2008) defines the New Double Difference Index, written as DDn (lower-case n), with the formulation:

DDn=2⋅ρλ−ρλ−Δ−ρλ+Δ,

for which we followed the common parameterization λ=710 nm, Δ=50 nm (i.e., 2R710−R660−R760). We confirm that our computations strictly used this DDn formula from [30]; the error was only in the index name.

To avoid confusion, we emphasize that this DDn is not the Normalized Difference Drought Index (NDDI), which is a drought metric defined as NDDI=(NDVI−NDWI)/(NDVI+NDWI) in the drought-monitoring literature. We did not use this drought index anywhere in our calculations; any appearance of “NDDI” in the earlier draft referred mistakenly to DDn and has now been corrected.

Corrections implemented:

  1. Table 1 redone: We have replaced the erroneous “NDDI” label with DDn (New Double Difference Index), provided the exact formula used and the correct citation to [30]. We also re-checked every other index in Table 1 to ensure that names, formulas, and references match the original sources. (No numerical results changed because our underlying computations already used the DDn formula.)
  2. Figure captions updated: All relevant figure captions now explicitly list the formula used for each index, as requested.
  3. Methods clarified: We added a brief note in Methods specifying the DDn parameterization (λ=710 nm,Δ=50 nm) and, where applicable, the use of the nearest sensor bands.
  4. Text cleanup: All occurrences of the ambiguous name have been corrected to DDn consistently throughout the manuscript.

We appreciate the reviewer’s careful reading; these corrections improve clarity and reproducibility without affecting our results or conclusions.

Second, again, the general order of the variables in a formula is not the formula for the vegetation indices (Table 2). This needs to be reformulated in the text and the table to make clear that the formula you used was "inspired" by the general formula of NDVI, etc. Thus, both Table 2 and Table 3 needs to be redone, and addition of the original reference is necessary. Specifically for table 3, how this you create those novel indices (line 174)?

Response: Thank you for pointing this out. We note that the variable ordering and related expressions of several formulas in Tables 2 and 3 of the original manuscript were not fully aligned with classical vegetation-index conventions, which could lead to confusion about their correspondence to standard indices. Accordingly, we have revised the main text and tables to clarify the provenance of each index (e.g., renaming DI and RI as DSI and RSI). All indices in Table 2 were computed strictly as defined in the cited references. We also state explicitly that the spectral-index formulas in Table 3 were inspired by canonical vegetation indices (e.g., NDVI) and were optimized using our correlation-matrix–based procedure. Tables 2 and 3 have been rebuilt to list, for each index, its name and explicit formula. In particular, for the new three-band (3D) spectral indices in Table 3, the Methods section now details their construction, including how our optimization algorithm generates and selects three-band combinations.

Third, as different wavelengths were used for the vegetation indices, the selection process of those wavelengths needs to be explained in the material and methods (section 2.3). 

Response: Thank you for pointing this out. In the revised manuscript, Section 2.3 (Materials and Methods) now explains the wavelength-selection procedure for each vegetation index. Specifically, we describe how correlation-matrix analysis was used to screen spectral bands: we first computed correlation matrices between winter wheat leaf nitrogen concentration (LNC) and individual bands as well as candidate band combinations, and then selected the bands/band sets with the largest correlations to LNC for index construction.

Fourth, for the model methods (section 2.4), the computational environment needs to be stated. Did you use some python or R libraires to do the calculation. Same apply to the creation of the figures, etc.

Response: Thank you for pointing this out. We have supplemented the information about the computing environment used for model training and graph drawing in the Materials and methods section of the revised manuscript. Our indices and models were calculated using MATLAB, and our graphs were created using Origin software.

Fifth, looking into the new Figure 2 and Figure 3, the selected combination of wavelength should be indicated in the figure. Furthermore, many of the vegetation indices seems similar. Those results needs to be discussed is the discussion, alongs the new Figure 4 results. What is the advantages of using the novel triplet indices, since Combine 5 is better?

Response: Thank you for pointing this out. Please refer to Tables 5 and 6 for the detailed wavelength-position combinations used in Figs. 2 and 3. Annotating these directly within the figures would obscure other graphical elements. The apparent similarity among indices stems from their shared operator forms (ratio/difference) and the same underlying data, which can produce similar visual patterns; however, the corresponding correlation coefficients differ substantially. Combination 5 integrates empirical indices with the three-band optimal spectral index (3D-OSI), thereby highlighting the advantages of 3D-OSI—particularly its superior estimation performance when combined with conventional empirical indices.

Sixth, reading the manuscript, its is difficult to undertand what is the Combine 7 (?). Combine 1 regroup all the variables from the index in Table 1, Combine 2 all the variables from Table 2, and Combine 3, all the variable from Table 3. Thus, combien 7 is all the vegetation indices?

Response: We thank the reviewers for noting that our descriptions of the input “combinations” were insufficiently clear in the original manuscript. In the revision, we explicitly define the meaning and composition of each combination (Combine X). Specifically: Combine 1 uses all traditional empirical spectral indices from Table 1 that are significantly correlated with LNC (p < 0.05) as model inputs; Combine 2 uses all two-band optimal spectral indices (2D-OSIs) from Table 2 with significant correlations to LNC (p < 0.05); Combine 3 uses all three-band optimal spectral indices (3D-OSIs) from Table 3 with significant correlations to LNC (p < 0.05). By extension, Combine 4 is the joint input of Combine 1 and Combine 2; Combine 5 is the joint input of Combine 1 and Combine 3; Combine 6 is the joint input of Combine 2 and Combine 3; and Combine 7 pools all indices from Tables 1–3—i.e., empirical + 2D-OSIs + 3D-OSIs—that are significantly correlated with LNC (p < 0.05), and uses them collectively as the model input set. These definitions have been added in Section 2.4 (Materials and Methods) to ensure unambiguous interpretation by readers.

Seventh, in the discussion, (line 339), you mentioned that the RF models With Combin 7 did not improve performance. You should mention relative to what: the ELM and BPNN models, or the overall predictive power of the model?

Response: Thank you for pointing this out. We note that the discussion in the original manuscript may not have clearly articulated the performance of Combine 7. In the revision, we have rewritten the relevant sentence to make the comparison baseline explicit. We now state in the Discussion: “When the Random Forest (RF) model uses Combine 7 (i.e., all indices) as inputs, its performance does not improve relative to combinations with fewer indices and even shows a slight decline.” We further clarify that this comparison is made against the best-performing combination within the RF model (e.g., Combine 5).

Eight, you mentioned in the discussion the methods SHAP. Add some referrences.

Overall, is still some work needed for this manuscript. Some rewriting of the different section might improve this manuscript. An overall results table for the different combination and single vegetation index might help the readers to better understand the conclusion of the manuscript.

Response: Thank you for pointing this out. In the Discussion, we describe using SHAP for model interpretation. Following your suggestion, the revised manuscript now includes the corresponding references supporting SHAP.

Comments on the Quality of English Language

Line 64: no hyphen between near and real-time.

Response: Thank you for pointing this out. We have revised this section.

Line 83. Missing a period between "combinations  Based"

Response: Thank you for pointing this out. We have revised this section.

Lines 83-87. This text is a little crude and needs reformulating.

Response: Thank you for pointing this out. We have rewritten the paragraph at lines 83–87 of the original manuscript to make the meaning clearer and the flow more natural.

Line 100-106 - new addition. This needs to be reformulated. It is unclear what are the objective: Create a correlation matrix?

Response: Thank you for pointing this out. We have supplemented the relevant paragraphs with an explicit statement of our research objective. For example, when introducing the correlation-matrix approach, we now add a sentence clarifying that its purpose is to screen optimal wavelength combinations for constructing new spectral indices. This makes it clear to readers that the correlation-matrix analysis is a methodological step serving subsequent index construction, rather than an end in itself.

There is still some capital P (P<0.05) in the text.

Response: Thank you for pointing this out. We have revised this section.

Line 169: NDVI is not in table 1.

Response: Thank you for pointing this out. We have revised this section.

Reviewer 3 Report

Comments and Suggestions for Authors

Complying with the reviewers' comments, the authors rendered this paper publishable as is.

Author Response

Reviewer3

Thank you to reviewer 3 for your affirmation of our work.

Round 3

Reviewer 2 Report

Comments and Suggestions for Authors

Dears authors,

The corrections made to the manuscript really improved its readability and it's now easier to understand the scientific method and the results presented.

There is still some minors corrections to the manuscript, since the modifications made introduced some additional questions.

First, its more clearer what is the DDn spectral indices. However, the definition in [30] is different from the one used un your study. In the original paper, the DDn is defined as: 

2 ⁎ ρλ1 − ρ(λ1 − λ2− ρ(λ1 + λ2)

where λ= 710 nm and λ2 = 50 nm, yielding the complete formula as 2 * 710 nm - 660 nm - 760 nm. Could you comment on the use of 2 * (710 nm - 2 * 760 nm) and indicates in the text why this modification was made ?

Some other minors corrections:

- Use the × symbol for the multiplication in the table instead of *

- For the Chlorophyll index, add space in the formula

- Line 22, remove [new] before DDn

- DDN spectral indice -> DDn (as per the original article)

- Line 100, remove (Geng et al. 2024) since the reference is already there [26]

- In the tables, Calculate Formula -> "Formulas"

- In the Table 1 captions: The [used] calculation formulas of empicical spectral index and [their] reference[s].

- Same apply to the other table captions.

- Line 203, [The  indices] i, j and k (in italics) represent ...

- Line 377-378, remove the parenthesis before "Compare) and after model

- References section: remove the doubling https://doi.org/ from the reference (e.g. see line 450)

Author Response

Comments and Suggestions for Authors

Dears authors,

The corrections made to the manuscript really improved its readability and it's now easier to understand the scientific method and the results presented.

Resource: We thank the reviewer for the careful reading and constructive suggestions. Below we respond point-by-point and list the exact revisions we will make in the manuscript.

There is still some minors corrections to the manuscript, since the modifications made introduced some additional questions.

First, its more clearer what is the DDn spectral indices. However, the definition in [30] is different from the one used un your study. In the original paper, the DDn is defined as: 

2 ⁎ ρλ1 − ρ(λ1 − λ2) − ρ(λ1 + λ2)

where λ= 710 nm and λ= 50 nm, yielding the complete formula as 2 * 710 nm - 660 nm - 760 nm. Could you comment on the use of 2 * (710 nm - 2 * 760 nm) and indicates in the text why this modification was made ?

Resource: Thank you for catching this. The displayed expression “2·(710 − 2·760)” in our text was a typographical error in the equation rendering, not a methodological change. In our analysis code and calculations we used the original Le Maire et al. (2008) definition of DDn:

DDn(λ1​,Δλ)=2R(λ1​)−R(λ1​−Δλ)−R(λ1​+Δλ),

and we followed the canonical setting λ1=710 nm, Δλ=50 nm, Δλ=50 nm, i.e., 2⋅R710−R660−R760 ​. We will correct the printed formula and the surrounding text to avoid any ambiguity. Because our code already used the original formula, all numerical results and conclusions remain unchanged.

Manuscript changes.

  • Replace the erroneous displayed formula with the exact definition above.
  • Add one clarifying sentence at first mention of DDn: “Following Le Maire et al. (2008), DDn was computed as 2⋅R710−R660−R760

Some other minors corrections:

- Use the × symbol for the multiplication in the table instead of *

Resource: Thank you for your correction, we have made the changes as requested.

- For the Chlorophyll index, add space in the formula

Resource: Thank you for your correction, we have made the changes as requested.

- Line 22, remove [new] before DDn

Resource: Thank you for your correction, we have made the changes as requested.

- DDN spectral indice -> DDn (as per the original article)

Resource: Thank you for your correction, we have made the changes as requested.

- Line 100, remove (Geng et al. 2024) since the reference is already there [26]

Resource: Thank you for your correction, we have made the changes as requested.

- In the tables, Calculate Formula -> "Formulas"

Resource: Thank you for your correction, we have made the changes as requested.

- In the Table 1 captions: The [used] calculation formulas of empicical spectral index and [their] reference[s].

Resource: Thank you for your correction, we have made the changes as requested.

- Same apply to the other table captions.

Resource: Thank you for your correction, we have made the changes as requested.

- Line 203, [The indices] i, j and k (in italics) represent ...

Resource: Thank you for your correction, we have made the changes as requested.

- Line 377-378, remove the parenthesis before "Compare) and after model

Resource: Thank you for your correction, we have made the changes as requested.

- References section: remove the doubling https://doi.org/ from the reference (e.g. see line 450)

Resource: Thank you for your correction, we have made the changes as requested.